# Precision medicine for liver tumours with quantitative MRI and whole genome sequencing (Precision1 trial): study protocol for observational cohort study

Fenella K Welsh,[1] John J Connell  ,[2] Matt Kelly,[2] Sarah Gooding,[3] Rajarshi Banerjee,[2] Myrddin Rees[1]

[1]Department of Hepatobiliary Surgery, Basingstoke and North Hampshire Hospital, Basingstoke, UK
[2]Perspectum, Oxford, UK
[3]Weatherall Institute of Molecular Medicine, University of Oxford, Oxford, UK

**Correspondence to**
Dr John J Connell;
john.connell@perspectum.com

## ABSTRACT

**Introduction** Radiogenomic analysis of patients being considered for liver resection is seldom performed in the clinic despite recent evidence indicating that quantitative MRI could improve posthepatectomy outcomes. Meanwhile, the increasingly accessible results from whole genome sequencing reporting on clinically actionable genetic biomarkers are yet to be fully integrated into the clinical care pathway.

**Methods and analysis** A prospective observational cohort study of up to 200 participants is planned, recruiting adults with primary or secondary liver cancer being considered for liver resection at Hampshire Hospitals NHS Foundation Trust. The data will be evaluated to address the primary endpoint to calculate the proportion of participants in which the results from whole genome sequencing would have resulted in a change in clinical management. Participants will be offered an additional non-invasive quantitative MRI scan prior to the operation and the impact of the imaging results on treatment decision-making will be evaluated.

**Ethics and dissemination** This study was reviewed by the NHS Health Research Authority and given favourable opinion by the Brighton and Sussex Research Ethics Committee (REC reference: 20/PR/0222). Research findings will be discussed with a patient and public involvement and engagement group, presented at relevant scientific conferences and published in open access journals.

**Trial registration number** NCT04597710

## Strengths and limitations of this study

► Precision1 will be the first study examining the clinical utility of radiogenomic information during the surgical treatment planning stage for patients with primary and secondary liver cancer.

► Our prospective study is designed to allow us to understand the extent to which quantitative MRI of the liver impacts surgical treatment planning.

► The additional value of whole genome sequencing will be evaluated to help us identify the presence of any clinically actionable genetic biomarkers which may support precision medicine provision to patients.

► All patients being considered for liver resection may be eligible for the study, regardless of tumour type or clinical background.

► The primary limitation of this study is the omission of a tightly matched negative control group where the impact of quantitative imaging on improving patient outcomes could be rigorously examined.

## INTRODUCTION

The incidence of treatable liver tumours is on the rise globally, driven by obesity, viral hepatitis and metastases from colorectal cancers. Survival rates can be improved with optimised allocation of treatment options including surgical resection, radiofrequency ablation, embolisation, chemotherapy and targeted molecular therapies (including immunotherapy). The key motivation of this study is to help patients access the most suitable treatment combinations by integrating radiogenomic[1] and clinical data.

A similar integrated approach, integrating radiology and pathology, has been shown to improve outcomes in breast cancer care.[2] Detailed pathological analysis of the surgical specimen from breast carcinoma biopsy provides valuable feedback to the radiologist, establishes the completeness of surgical intervention and generates predictive information for therapeutic decisions.[3]

Whole genome sequencing (WGS) has revealed cancer driver mutations and the complex molecular profile of liver cancer.[4] In many metastatic solid tumours, WGS has been used to identify a significant patient population (31%) who present with a biomarker that predicts sensitivity to a drug and lack any known resistance

biomarkers for the same drug.[5] Identifying which patients possess genomic variants that predict drug response or resistance will allow clinicians to make the optimal treatment decisions. The next technological challenge is integrating WGS into scalable clinical practice.

In addition to the molecular profile of the patient's liver cancer, a non-invasive assessment of the overall health of the liver using quantitative MRI will be carried out in this study. The MRI metrics from Liver-*MultiScan* can quantify changes to liver tissue characteristics, including fibro-inflammation and fat, with the accuracy and reproducibility of the results demonstrated in a number of studies.[6–8] A recent Innovate UK grant supported Perspectum, Hampshire Hospitals NHS Foundation Trust (HHFT) and University of Edinburgh to assess the use of quantitative MRI in decision support for liver resection. This prospective 143 patient study showed quantitative MRI using Hepatica (which incorporates Liver*MultiScan*) can predict length of patient stay.[9–11] Patients with liver cancer plus often undiagnosed chronic liver disease performed worse than those with healthy native livers; they also recovered more slowly, costing an extra 1.5 days of hospital stay per patient, a significant clinical impact.

The gold standard of liver assessment relies on imaging and biopsies. Advances in quantitative algorithms provide richer data about the cellular changes associated with liver disease, and greater insight into disease monitoring and treatment efficacy from MRI and digital pathology. In this project, by additionally analysing the genome of both tumour and germline, we aim to understand which variants are driving tumour growth and potentially amenable to targeted therapies and impacting treatment response, along with variants associated with response to surgery and liver regeneration.

In the previous work, we have demonstrated the ability to combine MRI, liver histopathology and circulating (enzyme) biomarkers to improve patient stratification and management.[7] The innovation in this project is to extend this technology to include discoveries from WGS and integrate them with quantitative MRI and histopathology data to produce a software product for clinical deployment to inform the management of patients with liver tumours. This approach would provide clinicians with a 'micro to macro' approach to cancer management, identifying genetic variants associated with tumour pathology, liver function and treatment efficacy, along with their association with pathological and radiological features that can be readily characterised in routine clinical practice. The complementarity of imaging with WGS will be captured by evaluating the current state of the liver from imaging and the future potential of the liver tissue to impact postoperative recovery such as genetic markers for reduced liver regeneration.

## METHODS AND ANALYSIS
### Study design
A prospective observational cohort study.

### Study objectives and endpoints
#### Primary objective
► To determine the utility of WGS to aid clinical decision-making in patients referred for liver resection.

#### Secondary objectives
► To determine the utility of quantitative multiparametric MRI with Liver*MultiScan* to aid clinical decision-making in patients referred for liver resection.
► To compare computationally derived digital pathology results with human pathologist assessments of the tumour and non-tumour tissues.
► To compare the histopathological assessments of liver fat and fibro-inflammation with quantitative MRI metrics (cT1, proton density fat fraction (PDFF)).
► To evaluate the long-term outcomes and recurrence rates and recurrence patterns of patients as it relates to imaging and WGS.
► To evaluate whether WGS enables better stratification of patients prior to surgery.

#### Primary endpoint
► Proportion of patients for whom clinically actionable data are provided by WGS at the time of surgery. This will be evaluated retrospectively, with clinically actionable data defined as data which would result in a clinician choosing a different medical intervention to the current standard of care.

#### Secondary endpoints
► Proportion of patients for whom clinically actionable data are provided by Liver*MultiScan*.
► Correlation of computationally derived digital pathology results with human pathologist assessments of the tumour and non-tumour tissues, along with assessment of intrarater and inter-rater variability.
► Correlation of MR measurements of steatosis and fibro-inflammation with digital pathology and human pathology.
► Performance of WGS and Liver*MultiScan* for predicting postsurgery length of stay in hospital, postoperative liver function, 1-year mortality and recurrence rates.
► Proportion of patients for whom actionable biomarkers of drug sensitivity are identified with WGS.

### Patient population
Patients who are due to undergo surgery for suspected liver tumour (primary or secondary). A sample size of up to 200 participants will be recruited from HHFT.

### Inclusion criteria
► Male or female 18 years of age and older willing and able to give informed consent to participate in the study.

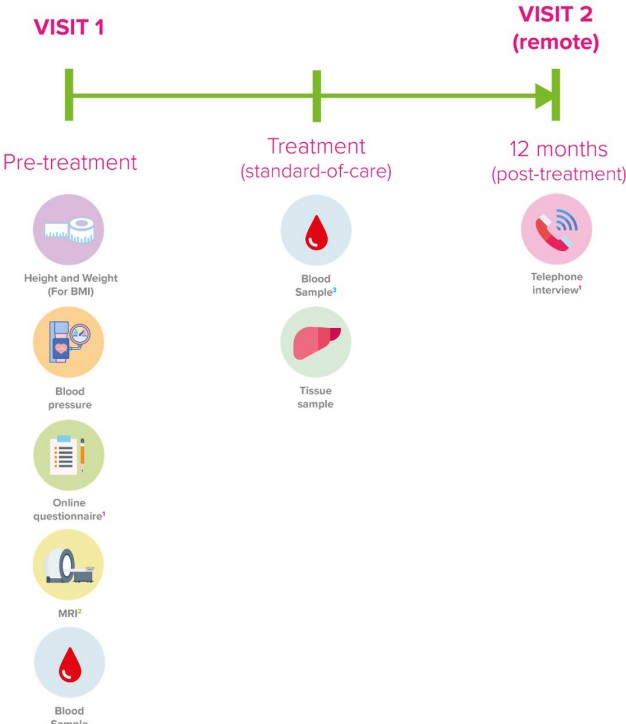

**VISIT 1**

**VISIT 2 (remote)**

Pre-treatment

Treatment (standard-of-care)

12 months (post-treatment)

Height and Weight (For BMI)

Blood Sample³

Telephone interview¹

Blood pressure

Tissue sample

Online questionnaire¹

MRI²

Blood Sample

**Figure 1** Study flow chart. Participants with informed consent will undergo pretreatment measurements at study visit 1 including height and weight, blood pressure, an online questionnaire, a non-contrast MRI scan and standard liver blood tests. Standard-of-care treatment will be followed and a blood sample and a sample tumour tissue explant will be collected. Twelve months following recruitment, a post-treatment telephone interview will occur as study visit 2. BMI, body mass index.

► Patients being considered for liver resection for primary or secondary liver cancer.

### Exclusion criteria
► The participant may not enter the study with any known contraindication to MRI (including but not limited to pregnancy, a pacemaker or other metallic unfixed implanted device, metallic fragments, extensive tattoos, severe claustrophobia).
► Any other cause, including a significant underlying disease or disorder which, in the opinion of the investigator, may put the participant at risk by participating in the study or limit the participant's ability to participate.

### STUDY ASSESSMENTS AND DATA COLLECTION
Study visits are summarised in figure 1.

### Clinical assessment
Minimal medical history data will be collected including information regarding previous diagnosis of liver cancer, cirrhosis and comorbidities including chronic kidney disease and diabetes. Information about alcohol consumption, smoking and coffee intake will be collected.

Information about any recent chemotherapy regimens will be collected.

### Blood tests
A blood sample will be collected as per routine clinical care, and blood results will be noted regarding biochemistry, renal profile, liver function tests, haematology, clotting, tumour markers including carcinoembryonic antigen, alpha-fetoprotein and CA19-9.

### Magnetic resonance imaging
Current routine clinical practice requests patients to visit the hospital 3 days prior to the planned surgery date for a COVID-19 test. At this visit, participants will be invited for an abdominal non-contrast enhanced MRI scan. Images will be processed to generate metrics indicating the liver tissue characteristics (Liver*MultiScan*) and future liver remnant volume (Hepatica) as previously described.[12]

### Impact of imaging on treatment decision-making
The clinical utility of quantitative MRI will be assessed through a survey of the consultant surgeons. The surgical plan prior to and after viewing the quantitative imaging reports will be noted, along with an indication as to the reason for any change in surgical plan with an expectation as to the impact on patient outcome:
► Improved health of residual liver.
► Probability of postoperative complications.
► Shorter recovery time.

### Tissue sample collection
During the surgical resection operation, a sample of tumour tissue from the explant will be stored in two manners: fresh frozen and formalin fixed/paraffin embedded. This tissue and a blood sample will be stored locally and then transported for tissue processing and WGS by an approved vendor.

### Whole genome sequencing
Genetic information from tumour tissue and from participant blood sample will be examined. A case review of all participants will be performed with these data to evaluate the clinically actionable information and what steps would have been taken by the clinical care team if this information were available prior to the operation.

### Digital pathology
Tissue explant samples will be analysed by the clinical histopathologist, and relevant information will be noted. Glass slides will be scanned and analysed using digital pathology methods to evaluate the objective scores of fibrosis, inflammation, ballooning and steatosis.

### Safety outcomes
No serious adverse events are anticipated as this study has been designed with minimal risks in mind, especially given the reduced healthcare capacity to deal with adverse events.

## Incidental findings

Any incidental findings will be reviewed by a radiologist and assessed by the clinical study team. If any structural organ abnormalities are found in the MRI scan or blood tests, a follow-up/note will be added to the participant's medical record as this may have implications for the participant. The participant can seek counselling/support for any potential abnormal findings from their primary healthcare provider should they wish to.

The chief investigator and/or principal investigator(s) will subsequently be pleased to discuss the medical findings directly with the primary healthcare provider, if requested.

## Statistical analysis plan

The data will be evaluated to address the primary endpoint to calculate the proportion of patients in which the results from WGS would have resulted in a change in clinical management. The change in intended management after WGS results will be modelled as a binary variable on the basis of a binomial distribution.

From the literature, we estimate a prevalence of actionable biomarkers derived from genetics of 31%. Using a 5% allowable margin of error and 95% CI, a sample of 200 is required to give a reliable sample to achieve a change in management of 25% of patients.

Descriptive statistics will be reported for all endpoints specified and any interim analyses. Normality of the data will be determined by use of the Shapiro-Wilk test and visual study of histograms. Results will be expressed as mean with SD for continuous normally distributed variables, as median and IQR for non-normally distributed data and as counts and/or percentages for categorical variables. Linearity of continuous data will be assessed by scatter plots. To explore any potential systematic biases, categorical and demographic characteristics expressed as number (%) will be compared using $\chi^2$. For all tests, a $p<0.05$ will indicate statistical significance.

Correlations between results from Liver*MultiScan*/Hepatica-derived quantitative MRI metrics and histological features, as well as results from Liver*MultiScan* and genomic features will be determined with Spearman's rank correlation coefficient and diagnostic performance will be assessed by the calculation of the receiver operating characteristic curve and the determination of the area under the receiver operating characteristic curve with 95% CIs. For all tests, a $p<0.05$ will be taken to indicate statistical significance.

To explore the impact of genetic variants associated with liver disease, the liver MR-derived biomarkers in those with and without known genetic variants associated with liver disease will be compared using a paired t-test (or non-parametric alternative).

Radiogenomic exploratory analysis will also be conducted to examine the correlations between image-derived biomarkers, genetic markers of organ damage or tumour malignancy. Digital pathology and circulation blood-derived biomarkers of the disease will be explored with Spearman's r (rs), with pairwise comparison conducted using two-sided Kolmogorov-Smirnov tests. Listwise deletion will be employed per analysis to included data sets with complete cases for all parameters in the model.

## Patient and public involvement

No formal patient advisory committee was set up and there was no patient or public involvement in the design and planning of the study. A patient and public involvement and engagement group has been assembled and will be consulted during dissemination of the findings.

**Collaborators** Precision1 Collaborators: Jo McClintock (Department of Hepatobiliary Surgery, Basingstoke and North Hampshire Hospital, Basingstoke, UK), Karen Scott (Department of Hepatobiliary Surgery, Basingstoke and North Hampshire Hospital, Basingstoke, UK), Rekha Rana (Department of Hepatobiliary Surgery, Basingstoke and North Hampshire Hospital, Basingstoke, UK), Hannah Stewart (Department of Hepatobiliary Surgery, Basingstoke and North Hampshire Hospital, Basingstoke, UK), Velko Tonev (Perspectum, Oxford, UK), Alessandro Fichera (Perspectum, Oxford, UK), Elena Bielli (Perspectum, Oxford, UK), Luis Núñez (Perspectum, Oxford, UK), Sina Knapp (Perspectum, Oxford, UK), Debbie Atkinson (Perspectum, Oxford, UK), Caitlin Langford (Perspectum, Oxford, UK), Soubera Rymell (Perspectum, Oxford, UK), J Michael Brady (Perspectum, Oxford, UK), Sarah Larkin (University of Oxford, Oxford, UK), Naser Ansari-Pour (University of Oxford, Oxford, UK), Mohammad Kazaroun (University of Oxford, Oxford, UK).

**Contributors** The study concept and design were conceived by FKSW, JJC, MK, SG, RB and MR. FKSW and MR will conduct the recruitment and data collection. Analysis and interpretation will be performed by FKSW, JJC, MK, SG, RB and MR. JJC prepared the first draft of the manuscript. All authors provided edits and critiqued the manuscript for intellectual content.

**Funding** This work was supported by Innovate UK (grant number: 50234).

**Competing interests** JJC, MK and RB work at and are shareholders of Perspectum.

**Patient and public involvement** Patients and/or the public were not involved in the design, or conduct, or reporting, or dissemination plans of this research.

**Patient consent for publication** Not required.

**Provenance and peer review** Not commissioned; externally peer reviewed.

**ORCID iD**
John J Connell http://orcid.org/0000-0002-7756-9997

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
