## [Reviewer comments · BMJ Open]

ARTICLE DETAILS

TITLE (PROVISIONAL)	Precision medicine for liver tumours with quantitative magnetic resonance imaging and whole genome sequencing (Precision1 trial): Study protocol for observational cohort study.
AUTHORS	Welsh, Fenella; Connell, John; Kelly, Matthew; Gooding, Sarah; Banerjee, Rajarshi; Rees, Myrddin

VERSION 1 – REVIEW

REVIEWER	Crispin-Ortuzar, Mireia University of Cambridge, Cancer Research UK Cambridge
REVIEW RETURNED	12-Nov-2021

GENERAL COMMENTS	-- General comments The manuscript presents the protocol for an observational cohort study focusing on radiogenomic data integration for liver cancer. With the advent of advanced computational integration techniques there is a growing need for more radiogenomic datasets, so the study is timely. The motivation and objectives are well described, but more details in the methods would be desirable to understand the complementarity with other studies and the potential for further work. -- Detailed comments * The timing of the different measurements is not entirely clear. For example, on page 8 line 51 it talks about the 'long term outcomes [...] as it relates to imaging'. Does that mean that follow-up imaging will also be part of the study? On page 13 line 19 it says that 'a blood sample will be collected' but it is not specified when. It would be useful to have a figure with the timeline. * It would be useful to have more details about the sampling process. Will only one sample be taken, or several? How will you choose the location (e.g. will you use an MRI co-registration technique?), and what will be the size of the tissue sample? * Can you give details about the MRI sequences that will be included? * What is the plan for data management? Where will the data be stored? The data sharing statement seems to imply that the dataset will not be made public. Is there a strategy in place to share the dataset with researchers who request it?
---

	* Page 15 line 58 talks about a prevalence of 31% - which biomarkers is this based on? Could you provide references? * Multiple tests are described in the statistical plan, appropriate multiple testing corrections should be applied.
REVIEWER	Morana, Giovanni Treviso Regional Hospital, Treviso IT, Radiology
REVIEW RETURNED	15-Nov-2021
GENERAL COMMENTS	the project is interesting and quite well supported by previous results. My main concern is that the number of patients could not be enough to reach results which can be clinically useful. My suggestion is to collect more cases using a multi center approach.

VERSION 1 – AUTHOR RESPONSE

Reviewer 1:

The manuscript presents the protocol for an observational cohort study focusing on radiogenomic data integration for liver cancer. With the advent of advanced computational integration techniques there is a growing need for more radiogenomic datasets, so the study is timely. The motivation and objectives are well described, but more details in the methods would be desirable to understand the complementarity with other studies and the potential for further work.

- The timing of the different measurements is not entirely clear. For example, on page 8 line 51 it talks about the 'long term outcomes [...] as it relates to imaging'. Does that mean that follow-up imaging will also be part of the study? On page 13 line 19 it says that 'a blood sample will be collected' but it is not specified when. It would be useful to have a figure with the timeline.

To aid in clarity of timings throughout the study, a figure has been added to the manuscript.

Secondary objective 4 states: "To evaluate the long term outcomes and recurrence rates and recurrence patterns of patients as it relates to imaging and whole genome sequencing." In the Precision1 study, participants will only be invited for a single MRI scan prior to their surgical operation, and no follow-up imaging will be included as part of the study; this will only be collected and used as clinically indicated. Long term outcomes and recurrence will be evaluated with respect to the imaging and WGS findings at the single point of collection.

The blood sample (for the purposes of WGS) will be collected from participants while they are being prepared for the operation as this can be collected through a cannula already placed for the purposes of the operation. This was designed to minimise the number of times a patient must be jabbed by a needle.

- It would be useful to have more details about the sampling process. Will only one sample be taken, or several? How will you choose the location (e.g. will you use an MRI co-registration technique?), and what will be the size of the tissue sample?

A tissue sample for Whole Genome Sequencing will be collected from the surgical explant. During hepatobiliary surgical operations, a research nurse is poised to collect a suitable piece of tissue from the explant for the purposes of clinical histopathological analysis that is routinely performed; an

additional sample will also be collected for the purposes of the research study. A typical sample is 1cm³. This has now been clarified in the manuscript.

- Can you give details about the MRI sequences that will be included?

We have added a citation to the relevant papers where the MRI sequences that are used have been previously detailed (Mojtahed et al., 2021).

- What is the plan for data management? Where will the data be stored? The data sharing statement seems to imply that the dataset will not be made public. Is there a strategy in place to share the dataset with researchers who request it?

Data and samples may be made available to support other ethically approved research in the future and this is explained within the Participant Information Leaflet detailed in a specific point within the Informed Consent Form.

We considered the full data management plan extraneous for this manuscript, but please find attached the full clinical trial protocol that has been reviewed by the Brighton & Sussex Research Ethics Committee under the NHS Health Research Authority. See page 21,22 detailing the data management plan. We have revised the manuscript to refer to this content within the 'Data sharing statement' section.

- Page 15 line 58 talks about a prevalence of 31% - which biomarkers is this based on? Could you provide references?

This statement cites a paper by Priestley et al., (Pan-cancer whole-genome analyses of metastatic solid tumours. Nature 2019). We refer specifically to a statement made at the top of page 215: "Half of the patients with a predicted candidate actionable event (31% of total) contained a biomarker with a predicted sensitivity to a drug ...". This is based on a vast analysis of genetic data from a range of tumour types and we feel fairly supports the statements we make in the Introduction of our manuscript.

- Multiple tests are described in the statistical plan, appropriate multiple testing corrections should be applied.

Multiple corrections methods will indeed be used during the statistical analysis of the data.

Reviewer 2:

- The project is interesting and quite well supported by previous results. My main concern is that the number of patients could not be enough to reach results which can be clinically useful. My suggestion is to collect more cases using a multi center approach.

We appreciate that the sample size of 200 may pose a risk of being too low to reach statistically significant results. As described in the statistical analysis plan, a power calculation was performed to estimate this sample size to achieve what was deemed a clinically useful change in management of

25% of patients. If our study in fact does not meet this 25%, a fair evaluation as to the clinical usefulness of MRI and whole genome sequencing will be performed and a larger sample size across multiple centres may well be pursued.

Editor's comments:

- Please revise the 'Strengths and limitations' section of your manuscript (after the abstract). This section should contain up to five short bullet points, no longer than one sentence each, that relate specifically to the methods. The results of the study should not be summarised here.

The Strengths and Limitations section has been revised.

- The name of co-author in ScholarOne system is different from the main document. Kindly amend accordingly.

ScholarOne system : Kelly, Matthew

Main document : Matt Kelly

This has been revised.

I hope that our responses and revisions to the manuscript are acceptable to the reviewers. Thank you for your time and consideration of our manuscript, we look forward to hearing from you.

VERSION 2 – REVIEW

REVIEWER	Crispin-Ortuzar, Mireia University of Cambridge, Cancer Research UK Cambridge
REVIEW RETURNED	09-Feb-2022
GENERAL COMMENTS	The authors have addressed my comments and I have no further concerns.